# Effect of Defocused Nanosecond Laser Paint Removal on Mild Steel Substrate in Ambient Atmosphere

**DOI:** 10.3390/ma14205969

**Published:** 2021-10-11

**Authors:** Zhong Zheng, Chaofan Wang, Gang Huang, Wenju Feng, Dun Liu

**Affiliations:** School of Mechanical Engineering, Hubei University of Technology, Wuhan 430068, China; 1923536180@139.com (C.W.); huanggang1008@hotmail.com (G.H.); 13460192991@163.com (W.F.); dun.liu@hbut.edu.cn (D.L.)

**Keywords:** laser cleaning, defocusing, compressed air, inert atmosphere, DC01 steel

## Abstract

The obvious advantages of laser paint removal technology make it a viable alternative to traditional paint removal methods. Infrared nanosecond laser was used to remove paint from car body. The microstructure, composition, surface roughness, hardness and ablative products of the samples were analyzed. The effect of the process combination of laser defocus distance and ambient atmosphere (ambient air, compressed air and inert atmosphere) on the substrate damage and the paint removal effectiveness was explored, and the related mechanism was discussed. Defocus not only changed the fluence of laser spot but also increased the spot diameter. The effect of defocused laser paint removal on the paint and substrate was caused by the superposition of these two factors. The results show that the laser with defocus distance of +4 mm effectively removed the paint in inert atmosphere and has the least adverse effect on the substrate. The content of C element and organic components on the substrate surface was the lowest, and its surface roughness and hardness was very close to the uncoated substrate. Focused laser paint removal in ambient air caused the most serious damage to the substrate. Its surface microhardness increased by 11 HV, and the influence depth reached 37 µm. The mechanism of laser paint removal without auxiliary gas is the superposition of laser plasma effect, laser gasification effect and thermal stress effect. In open atmosphere (compressed air and inert atmosphere), the mechanism of laser paint removal is laser gasification effect and thermal stress effect. This research can provide practical references and theoretical basis for the large-scale industrial application of low/non-damage laser paint removal technology.

## 1. Introduction

The surface of a vehicle is painted to avoid body corrosion, decorate and distinguish between different uses. The repair, removal and replacement of the paint on the car body are an important part of car maintenance, including peeling off the paint in specific areas to expose the substrate.

Compared to the above paint removal methods, laser shows obvious advantages [1], making it a viable alternative to traditional paint removal methods: (1) non-contact depaint, so the substrate is not subject to external mechanical force, damage and chemical pollution; (2) high accuracy and selectivity; (3) controllable and easy to achieve automation; (4) the solid waste is much less than the initial amount of paint and is also green and environmentally friendly. These advantages are particularly important in applications such as high-end vehicles, rail vehicles and military vehicles where substrate integrity is more strictly required, and it is necessary to ensure that the paint is removed accurately and completely, while the adverse effects on the substrate are negligible.

Since A.L. Schawlow [2] first proposed the “laser eraser” to vaporize ink without ruining the paper in 1965, and S.M. Bedair [3] determined the concept of laser cleaning in 1969, great progress has been made in basic research and development of supporting equipment for laser cleaning. Nowadays, it has been successfully applied in military aerospace and nuclear power and other cutting-edge or special fields [4,5].

Laser paint removal on metal substrates has been studied extensively. In view of the characteristics of laser itself, such as the small spot size, the paint removal efficiency in industrial applications is one of the focuses of users’ attention. Therefore, a considerable number of studies have aimed at the efficiency, effectiveness and feasibility of laser depaints technology [6], including the relationship between the laser depainting process and paint removal rate (surface cleanliness), removal efficiency and repainted paint adhesion [7,8,9]. For example, Razab, M.K.A.A et al. [5] reviewed the potential and mechanism of pulsed Nd: YAG laser for coating removal in automotive industry. Li, X et al. [10] used a 1064 nm Nd: YAG nanosecond pulse laser to remove paint from the surface of Marine steel structures. The influences of three groups of laser parameters (Average power, Repetition rate and Scan speed) on surface cleanliness, surface roughness and adhesion of recoating of steel structures were studied. The results showed that the surface treatment standard of SA2. An excellent bonding strength of 20 MPa was achieved between the recoated coating and the substrate, higher than the requirements of the shipyard. Kuang, Z et al. [11] used a 1064 nm nanosecond pulse laser to remove paint from the surface of aluminum alloy sheets used for rail vehicles. By defocusing the laser and blowing inert atmosphere (argon) into the treated area, sparks and combustion flames were effectively suppressed. The maximum paint removal rate was 0.06 m^2^/min. Surface melting changed the surface roughness of the substrate when it absorbed the extra energy of the laser beam. In Chen G X et al. [12], the cleaning process was carried out with a single sealed CO2 laser (Rofin-Baasel, Multiscan VS). The influence of laser parameters (laser power density and overlapping of neighboring scan lines) on surface quality and depaint speed was studied. The results showed that Nd: YAG laser ablation provides complete cleaning with high surface roughness due to substrate damage. They also used Raman spectroscopy to track the amorphous carbon on the treated samples and determined that the mechanism of laser paint removal was thermal decomposition.

In addition, some scholars have conducted in-depth studies on the mechanism of laser paint removal on metal substrate surface [13]. For example, Jinghua Han et al. [14] used a 1064 nm nanosecond pulse laser to remove epoxy polyester mixed paint from the surface of aluminum substrate. The results showed that laser paint removal is mainly affected by laser vaporization effect, thermal stress effect and laser plasma effect, and the thermal stress effect is the largest. It also has the best paint removal effect and can avoid damage to the substrate caused by vaporization and plasma effect. Zhao Hai Chao et al. [15] used a 1064 nm pulsed laser to remove 50 μm thick polyacrylic resin primer paint from the surface of an aircraft skin (LY12 aluminum alloy plate). The influence of process parameters such as scanning speed, pulse frequency, scanning line interval and laser power on paint layer stripping was studied. By analyzing the surface morphology and the particles collected during depainting, three possible mechanisms of paint removal were proposed: combustion reaction, thermal stress vibration effect and plasma shock effect.

The influence of laser paint removal process on metal substrate has also been studied [16,17,18]. For instance, Shamsujjoha M et al. [19,20] used a 1064 nm Nd: YAG nanosecond laser pulse to remove the red epoxy paint from a high strength steel used in shipbuilding. The effects of laser parameters on surface roughness, microstructure, hardness, repainting adhesion, residual stress state and fatigue properties of the underlying substrate material (carbon steel) were studied. The results showed that the underlying metal substrate was melted and re-solidified. The melting depth was 1–5 μm. The surface appearance had visibly changed. The adhesion of the repainted samples subjected to laser ablation coating removal (LACR) was comparable or even superior to that of the abrasive blasted and repainted samples. The residual stress was typically 242 ± 63 MPa in tension and confined to the shallow depth of ~35 μm. Fatigue tests confirmed that the LACR-treated samples performed just as well as abrasive blasted samples. Guodong Zhu [21] et al. used an Nd: YAG laser with 1064 nm wavelength to remove BMS10-11 paints from the surface of the Boeing series aircraft skins. The effects of laser energy density on the surface depainting effect, surface morphology, friction and wear properties, microhardness and residual stress and corrosion performance were investigated. The results showed that compared with mechanical lapping, laser lapping did not reduce the corrosion resistance and friction and wear performance of aircraft skin, whereas after laser depainting, the aircraft skin surface produced a certain plastic deformation and hardened, increasing the residual tensile stress.

According to the published literature, the current laser paint removal process cannot completely avoid the damage to the substrate, and the theoretical research is not systematic and thorough enough yet. The theoretical research on the effect of defocusing and auxiliary gas on the metal substrate and paint layer is not systematic and in-depth. This research aims to explore the laser paint removal process and related paint removal mechanism for the non-destructive cleaning of car body substrates by the use of defocus and auxiliary gas. Infrared nanosecond fiber laser is used to remove paint on the car body surface. By analyzing the microstructure, composition, surface roughness, hardness and ablative products of the samples, the effects of laser defocus distance and ambient atmosphere (ambient air, compressed air and inert atmosphere) on the effectiveness and efficiency of paint removal and the substrate damage were explored. The feasibility and related mechanism of laser paint removal on the car body were discussed. The study can provide practical reference and theoretical basis for the large-scale industrial application of low/no damage laser paint removal technology.

## 2. Materials and Methods

### 2.1. Raw Materials

The samples with four-layer paint used in this study were cut from automobile body coverings made in China, as shown in Figure 1. Figure 1b shows the top-view of the sample. Figure 1c shows its cross-section, where the colors and thicknesses of each layer are indicated, and the total thickness of the four-layer paint is about 160 µm ± 10 μm. Paints are mixtures of resins, solvents, pigments and additives, as shown in Table 1. The bottom layer of Figure 1c is the substrate. The material is DC01 cold-rolled low-carbon steel (carbon content ≤ 0.12 wt%, melting point 1455 °C, thickness 1.8 mm). Corresponding to the central rectangular area 1 of Figure 1b, this is the exposed substrate after all the paint layers are removed by the laser, showing a bright metallic background color. In Figure 1c, there is a primer layer on top of the substrate, and the material is epoxy paint, which corresponds to area 2 in Figure 1b. The remaining 3 layers in Figure 1c are all acrylic modified resin paints. From bottom to top, they are the midway layer (corresponding to area 3 in Figure 1b), the colored paint layer and the varnish layer (corresponding to Figure 1b, Area 4).

### 2.2. Laser Paint Removal

The schematic of the setup for laser paint removal is presented in Figure 2. The main parameters of the nanosecond pulsed laser systems used (SP-100W-EP-Z, SPI Lasers UK Ltd. Southampton, UK) are shown in Table 2. The samples were placed on the processing platform. The laser beam was perpendicular to the sample surface. The laser treated area for each condition was 10 mm × 10 mm. The defocus distance refers to the distance between the samples surface and the focal plane of the laser beam, which was controlled by adjusting the focal plane position along the optical axis (Z axis). The laser F-θ scanning lens (lens focal length f = 254 mm) is mounted on the Z axis, which can be moved up and down manually, so that the laser beam focal plane can be moved along the optical axis. The laser beam scans along the X and Y directions. In order to ensure uniform energy coverage on the laser-treated surface, the spot overlap rates were always consistent in the X and Y directions during the laser treatment, and the Zigzag scanning method was adopted, as shown in Figure 3. Silicon wafers (20 mm × 20 mm × 1 mm) were placed 8 mm above the laser-treating area to collect sputtering and vaporized products.

The gas was blown to the laser-treating area in an open environment. The gas flow was controlled within 8–10 Lpm through the pressure gauge. The distance between the air outlet and the laser-treating surface was 15 mm. The angle between the airflow and the laser-treating surface is 30°. The purity of inert gas is 99.5%.

Based on this, many exploratory experiments were carried out by changing the laser process parameters (pulse repetition frequency, scanning speed, spot overlap rate, energy density and laser scanning path) to optimize the parameters for the paint removal. These preliminary test results were visually evaluated under an optical microscope, and the optimized parameters of the focused laser paint removal were determined by taking the efficiency and effect of paint removal into consideration, as shown in Table 3. Due to the thickness of the four-layer paint (about 160 µm ± 10 μm), the 100 W nanosecond pulsed laser used in this work needs to scan three times to obtain a good removal effectiveness.

The laser spot diameters at different defocus planes are calculated based on the far-field divergence angle and asymptote, as shown in Table 4 and Figure 4. The relationship between the average laser energy density and the laser spot size can be described by Equation (1) [22]:(1)$$E_{f}=\frac{Q}{\pi \times W_{f}^{2}}$$
where, $E_{f}\;$ is the average laser fluence, *Q* is the energy of the laser, and $W_{f}\;$ is the radius of the laser beam.

The schematic diagram of laser spot, focal length and focusing amount is shown in Figure 5. Then the relationship between the spot diameter d, the focal length λ and the defocus amount L acting on the sample surface can be expressed as
(2)$$d=\frac{L\left(D_{0}-D\right)}{\lambda }+D$$

It can be seen from Table 4 that defocusing changes the energy density of the laser spot radiated on the surface of the material and the radiation diameter of the spot. The defocused spot is the combined result of the two superimposed effects. Therefore, it cannot be obtained simply by optimizing the laser intensity at a focal point.

### 2.3. Characterisation and Tests

Optical microscope (Nikon, Tokyo, Japan) was used for metallographic observation. The samples were cut with a band saw. The cross-sections of the samples were ground with SiC paper as low as 2500 grade and then polished with diamond paste as low as 0.25 μm. They were then etched in a solution (3% nitric acid in ethanol) for 45 s to reveal the microstructure.

Ultra-high resolution field emission scanning electron microscope (FE-SEM, SU8010, Hitachi, Tokyo, Japan, equipped with energy dispersive spectrometer (EDS)) was used to characterize the microstructure. The chemical composition was determined by X-ray photo-electron spectroscopy (PHI 5000 Versa Probe Ⅲ UIVAC-PHI Company, Chigasaki, Japan). An ultraviolet-visible spectrophotometer (Lambda750S, Perkin Elmer Co., Ltd., Waltham, MA, USA) was used to measure the absorbance of the samples in the ultraviolet-visible-infrared range (300–1800 nm). The linear roughness Ra of the sample surface was obtained by means of a three-dimensional surface white light interferometer (Contour GT-K0, Bruker, Germany), and the measured data of five different parts of each sample surface were averaged.

An automatic micro-Vickers hardness tester (HMV-G21, Shimadzu, Kyoto, Japan) was used to measure the microhardness of the cross-sections of the polished samples in ambient air at room temperature. The load was 200 g, and the loading time was 15 s. The measuring points were arranged along the processing depth direction. The first measuring point was 5 μm away from the processing surface along the depth direction, and the distance between each measuring point after the second measuring point was 8 μm along the depth direction. There are 5 depth measurement positions, and the average of the measurements from 10 different positions at the same depth was taken.

## 3. Results and Discussion

### 3.1. Macroscopic Phenomenon

Table 5 records the phenomena that occurred during each group of laser paint removal experiments. Figure 6 is a schematic diagram of laser paint removal at the focal point in the ambient air, which produces plasma, flames, flying debris and smoke. During the focused laser paint removal in inert atmosphere (Ar), there are no plasma, flame, debris splash, smoke and other phenomena but only bright and dazzling spots. During the defocused (defocus distance +6 mm) laser paint removal in ambient air, there are no plasma, debris splash, smoke and other phenomena, but flame combustion is visible. During the defocused (defocus distance 0–+4 mm) laser paint removal in ambient air, plasma, flame, debris splash, and smoke occurred. Based on previous experience and published literature studies, we concluded that ionization occurred in the lacquer layer, and the removal mechanism of paint was mainly laser plasma effect [23,24]. In other words, laser with high energy density generated free electrons in the paint layers through the multi-photon effect or thermal electron emission. Free electrons, as seed electrons, continuously absorbed laser energy through inverse bremsstrahlung radiation, thus forming a high-density laser plasma [25]. The plasma had temperatures of more than a dozen eV and could radiate UV and soft X-rays, which were absorbed by the paint layer, further ionizing and heating the paint layer [26]. The plasma also expanded, forming a high-pressure shock wave to remove the paint layer by sputtering [27,28]. In addition, the laser vaporization effect and thermal stress effect had little influence, but they all contributed to the violent chemical reaction and physical change. 

During the defocused (defocus distance +5–+7 mm) laser paint removal in ambient air, there was only flame. At this time, the energy density of the spot was lower than that of the defocused (defocus distance 0–+4 mm) laser paint removal. It can be seen that the plasma can be significantly suppressed by reducing the laser energy density to the ionization threshold of the processed materials. During the laser paint removal in open atmosphere (compressed air and inert gas, gas flow 8–10 Lpm), whether defocus or not, there was no plasma, flame, debris splash, smoke, flame, etc.; only bright and dazzling spots were seen. According to the preliminary judgment, the main mechanism involved in this situation is the laser vaporization effect or thermal stress effect [29].

Figure 7 shows the photos of the samples after laser scanning once in ambient air. It can be found that one layer of paint can be completely removed by laser when its defocus distance was 0–+2 mm, and there are gray-white particles deposited in and around the scanning area. With the increase of defocus distance, the amount of paint removed by laser decreased gradually. This is because the energy density of laser spot decreases with the increase of defocus distance; the influence of laser plasma effect on paint removal decreased, and that of laser vaporization effect and thermal stress effect increased. The top layer of transparent varnish was removed by laser when its defocus distance was +3–+7 mm; the blue paint layer and intermediate paint layer were peeled off. In this case, due to the decrease of the energy density of the spot, the presequence laser pulse only warmed the paint layer, and the subsequent pulse continued its transformation into heat energy to ablate and vaporize the paint. In addition, the elastic thermal vibration produced by light waves can also destroy the adhesion between the paint layers, that is, the van der Waals forces or hydrogen bonds between the polymer chains, leading to the blue paint and intermediate paint to exfoliate due to vibration [30].

### 3.2. Paint Removal Effect

Figure 8 shows the photos of the samples laser scanned 3 times in ambient air and in inert atmosphere (Ar). Based on the laser parameters used in this experiment, the paint cannot be completely removed by laser when its defocus distance was greater than +5 mm, so it is not be shown here. It can be seen that the metallic luster of the samples surface after the focused laser paint removal was brighter than that of the unpainted substrate. With the increase of defocusing distance, the metallic luster on the samples gradually darkened. The appearance of samples laser treated in inert atmosphere (defocus distance +4 mm) was very close to the unpainted substrate, and no paint residue could be found with naked eyes. Some paint remains on the samples laser treated in ambient air (defocus distance +4 mm). This is because when the defocus distance was +4 mm, the energy density of laser spot decreased. In addition, the plasma, flame and smoke generated by laser paint removal in ambient air shield part of the laser energy, resulting in the paint cannot be completely removed. During the laser paint removal in inert atmosphere, the airflow reduced local heat and blew away the smoke and ablation particles generated. Free electrons could not accumulate, which weakened the laser plasma generation conditions, and then avoided the substrate damage by violent reaction. Because there was no plasma, flame, smoke and other shielding laser beam, laser energy could act fully on the paint. Furthermore, the blowing force of strong air was equivalent to exerting a certain external force on paint layer, which also accelerated the removal of paint.

Figure 9 shows relative amount of C-C/C-H functional groups on samples measured by XPS. The content of C-C/C-H, C-O, COO^−^, C=O on the samples laser treated in compressed air and in inert atmosphere changed compared with that of the unpainted substrate. The C-C/C-H content represents organic components, including residual oil, laser ablation products such as carbides and paint residual organic carbon on samples. The results show that the relative content of C-C/C-H bonds in the samples was more than 81% (the same goes for unpainted substrate). The organic components detected on the unpainted substrates came from trace oil stains. The content of organic components in the samples after focused laser paint removal was the highest. That in the samples after laser paint removal in inert atmosphere (defocus distance +4 mm) was the lowest, which was reduced by 1.88% compared with the unpainted substrate. Therefore, the effectiveness of laser paint removal in inert atmosphere (defocus distance +4 mm) was optimal.

### 3.3. Surface Absorptivity

Figure 10 shows DC01 steel substrate measured by UV-visible spectrophotometer in the UV-visible-infrared range (300–1800 nm). Figure 10 and Table 6 shows the absorbance of samples laser treated in inert atmosphere with different defocus distances and unpainted substrate. The laser scanned 3 passes when the defocus distance was ≤+4 mm, and it scanned 4 passes when the defocus distance was +5 mm. The absorbance of the laser treated samples changed to varying degrees. That of the laser treated samples (defocus distance ≤ +3 mm) decreased. Curve (5) in Figure 10 shows that the absorbance of the laser treated samples (defocus distance +4 mm) was very close to that of unpainted substrate (Curve (7) in Figure 10). This is mainly because the surface roughness and composition of the laser treated samples changed to different degrees. The reflection and refraction behavior of incident light change with the surface roughness, which leads to the change of absorbance. The absorbance of the laser treated samples (defocus distance +5 mm) increased greatly, and it can be seen that there was a blue-black film on the samples, showing typical macroscopic characteristics of steel oxidation. This is because the samples underwent one more laser scan passes than the other samples, and the metal substrate was directly exposed to the high temperature irradiation of the high-energy laser beam for a longer time, thus accelerating the oxidation and discoloration. As there was paint residue on the laser treated samples (defocus distance ≥ +6 mm), it is not listed here.

### 3.4. Characterization of Substrate

Figure 11 shows the micro morphology of the sample surface and cross section after laser paint removal. In the ambient air and in an inert atmosphere (Ar), the top-views of the sample after the paint removal when the defocus is 0 mm are as shown in Figure 11a–d. Both show pits and ridges that flow and resolidify after the metal melts and are similar in size. However, the top views of the two samples are not quite the same. After laser paint removal in ambient air, the surface of the sample (Figure 11e) has a layer of loose, rough and uneven remelted material, with large undulations in the depth direction and uneven thickness, about 8–11 μm. The magnified SEM image (Figure 11e-1) shows that it has formed an obvious grain refinement phenomenon. Because the laser processing speed is very fast, the heating speed and quenching effect lead to the formation of martensite, especially its grain refinement and the uniformity of the hardened zone. Martensitic or austenitic stainless steels produce fine structures at high cooling rates. In laser transformation hardening, the cooling rate usually exceeds 1000 °C/s, the surface structure is martensite, and the grain size is reduced to about 200 nm.

In the inert atmosphere, the remelted substance on the surface of the sample (Figure 11g) after laser paint removal has small fluctuations in the depth direction, and the thickness is much thinner, about 3–5 μm. The magnified SEM image (Figure 11f-1) shows that it has also formed the phenomenon of grain refinement. The paint removal process in the ambient air is mainly affected by the laser plasma effect. Due to the high energy density of the laser spot at the focal point, after the substrate absorbs the laser energy, the temperature rapidly rises to the melting point and melts. The high-pressure shock wave generated by the plasma plume pulls part of the molten base metal out of the molten pool. The ejected molten metal falls back to the surface of the sample and rapidly cools to form the remelting layer in the Figure 11e [31]. The remaining molten metal in the molten pool is squeezed to the edge of the molten pool by light pressure and vaporization pressure. After the pulse, it quickly cools and solidifies, forming ablation pits and ridges. In the process of paint removal in an inert atmosphere, no high-temperature and high-pressure plasma are generated, and no violent physical and chemical phenomena such as splashing occur. Only the molten metal is squeezed to the edge of the molten pool by light pressure and gas pressure, and after cooling and solidification, shallow pits and ridges are formed.

The micro morphologies of the samples laser treated (defocus distance +4 mm) in compressed air and in inert atmosphere are shown in Figure 11g–j. Both are almost identical to the unpainted substrate. The substrate has no remelting and solidification phenomenon, no ablation pits, etc.

Figure 12 shows the energy dispersive X-ray Spectroscopy (EDS) spectrum of the unpainted substrate and samples laser treated in ambient atmosphere. Compared with the unpainted substrate, the content of elements C and O in the samples laser treated in compressed air and in an inert atmosphere increased slightly. The content of C in the focused samples laser treated in compressed air was the highest, which increased by 10.76%. That in the samples laser treated (defocus distance +4 mm) in inert atmosphere was the lowest, only increased by 2.4%. The reason is that C-H, C-C, C=O and other chemical bonds were broken and rearranged in the polymer molecular chain of acrylic resin and epoxy paint under the action of chemical bond breaking and combustion mechanism [32] during laser paint removal in compressed air and inert atmosphere. Ablation produced amorphous carbon, carbides and paint residual organic carbon, in which larger particles and smoke were blown away by strong airflow, and some micro/nano-particles attached to the samples at high temperature. The energy density of focused laser spots was high, and the substrate metal melted and then re-solidified, so there were more micro/nano-particles attached, and more carbon migrated into the substrate. The reason for the lower C content in the samples laser treated in inert atmosphere than that in compressed air may be that the paint ablates more fully in compressed air, resulting in more amorphous carbons and carbides. The content of organic components in the samples laser treated in compressed air and inert atmosphere was lower than that of the unpainted substrate, which may be due to the removal of trace oil on the original substrate.

### 3.5. Roughness

The reduced surface roughness may reduce the adhesion of the subsequent painting and may not meet the requirements of the re-painting specification. If the surface roughness is too high, the paint will be wasted, and the air may be trapped in troughs and cause bubble defects. Figure 13 shows the surface roughness corresponding to the photos of the samples laser scanned 3 times in Figure 9. Because of the residual paint on the samples laser treated (defocus distance +4 mm) in ambient air, the surface roughness of the samples cannot be accurately displayed, so it is not displayed here. The surface roughness of the samples laser treated (defocus distance +4 mm) in compressed air and inert atmosphere was very close to that of unpainted substrate. That of the focused laser treated samples increased the most. They increased gradually with the reduction of the laser defocus distance.

### 3.6. Microhardness

Figure 14 shows the microhardness of the unpainted substrate and that of the laser treated samples at different depths from the treated surface. Compared with the unpainted substrate, the microhardness of laser treated samples increased near the treated surface (5 μm away from the treated surface). The microhardness of the samples focused laser treated and the samples laser treated (defocus distance +3 mm) in ambient air was the largest (more than 11 HV), and gradually decreased until a depth of 37 μm with the increase of depth. However, the microhardness of the samples laser treated (defocus distance +4 mm) in inert atmosphere did not significantly increase. This is because the metal on the samples focused laser treated melted and re-solidified to form a 5–11 µm remelting layer (Figure 11e,f). As the laser pulse action time was very short (ns), both metal melting and resolidification occurred in a very short time, which is equivalent to surface quenching. The resolidification layer was martensite, which is hard. Meanwhile, a high-density plasma jet explosion was generated during laser paint removal, and the peak stress of the impact stress wave (high pressure shock wave) was much higher than the dynamic yield strength of the substrate material [33], leading to work hardening [34,35] of the material and increasing the microhardness. This process is equivalent to laser shock strengthening, resulting in residual compressive stress deep into a depth of 37 µm. However, there is no obvious remelting layer on the samples laser treated (defocus distance +4 mm) in inert atmosphere and no plasma generated during the process of paint removal. As a result, there was little change in microhardness compared to the unpainted substrate. In addition, the increase in carbon content may also lead to an increase in surface hardness. Compared with the unpainted substrate, the content of C element on the laser treated surfaces increased (Figure 12). Because of the chemical potential gradient, it can be assumed that alloying elements were redistributed in the steel through diffusion processes. The atomic ratio of iron to carbon is about 0.2, and the diffusing element carbon is much smaller than the steel matrix unit, so carbon can move between the iron matrix vacancies, resulting in interstitial diffusion [36].

### 3.7. Characterization of Particles Collected on Silicon Wafers

A mixture of particles from several microns to several hundred microns, vapors, clusters and ionized nanoclusters generated during the process of evaporation or ionization of pain [37,38], which diffused outwards and deposited on the surface of the silicon wafer [39], as shown in Figure 15. The collected particles, ranging in size from micrometers to nanometers, can be seen in a variety of shapes, including flakes, clumps, and spheres. Figure 16 shows the energy dispersive X-ray Spectroscopy (EDS) spectra of Figure 15a–m. Figure 15a–f represents the particles collected during laser paint removal (in ambient air, defocus distance 0 mm), and the EDS spectra of Figure 16a–g corresponds to it. In Figure 15a surfaces are smooth and complete with clear and flat edges, and no signs of melting or ablation are found, indicating that they had cracked before melting. EDS analysis showed that their content of C and O was no different from that of the original paint, which could be judged as the paint fragments peeled off due to the thermal stress effect. Those are paint pieces that peeled away from the cracks due to thermal stress effects. The micron agglomerates in Figure 15b have irregular shapes and smooth surfaces. EDS showed that their C content decreased a little and oxygen increased a little. They were identified as melted paint particles. Figure 15c shows the micro-clusters composed of several paint nanoparticles, whose C content significantly decreased and oxygen greatly increased. Those are clusters of paint nanoparticles generated by thermal ablation that attract, combine and aggregate with each other. The content of C of paint microspheres in Figure 15d decreases sharply, while their content of oxygen increases sharply. They were produced by ablation or ionization [40,41]. Nano-spheres in Figure 15f were formed by condensing gaseous molecules formed by the paint decomposition and combustion on the surface of silicon wafers. EDS spectra of the black pellets in Figure 15e showed that the main component was Fe, in addition to 29.42% oxygen, and the content of C was very small. They were identified as small iron balls condensed after the substrate metal melted and splashed out. This is evidence of substrate damage.

Figure 15i–l shows the particles collected during focused laser paint removal in inert atmosphere. Most of them are flakes or spherical particles, which are mostly the result of chemical bond breakage caused by photo-thermal ablation of the paint [32], and no micron agglomerates similar to those produced by paint melting in Figure 15b are found.

Figure 15n shows the micron paint flakes collected during laser paint removal (in inert atmosphere, defocus distance +4 mm). EDS analysis showed that their content of C and O was no different from that of the original paint, which could be judged as the paint fragments peeled off due to thermal stress effect. Local heat absorption caused the paint film to tense and buckle and then cracks to form around the bending areas, which subsequently caused the affected parts of the film to peel off. No small black iron ball was found in the particles collected during laser paint removal in inert atmosphere.

Figure 17 compares the contents of C, O and Fe elements of primer particles collected by silicon wafers. Figure 15g shows micron agglomerates of primer particles collected during focused laser paint removal in ambient air, with irregular shape, and two parts with distinct morphologies can be seen. The lower left half has a rough surface with small cracks; the lower right half has a smooth surface, similar to Figure 15b. The EDS spectra of Figure 16g1 and Figure 16g2, respectively, shows that iron was detected in its lower left part, with more oxygen content and less C content, which was judged to be the substance separated from the substrate. No iron was detected in the lower right part, and the contents of C and O were similar to those in Figure 15b, which was judged to be epoxy primer with incomplete ablation. Figure 15h shows micron primer agglomerates collected during laser paint removal (in ambient air, defocus distance +3 mm) without the cracks shown in Figure 15g. Compared with the primer particles collected during focused laser paint removal in ambient air (Figure 16g1), the content of iron and oxygen was reduced, and the content of C was increased. Figure 13i shows micron primer flakes collected during focused laser paint removal in inert atmosphere, flat and thick, without cracks on the surface. Compared to the primer particles collected during focused laser paint removal in ambient air (Figure 16g1), the content of iron and oxygen was reduced more, the content of iron was reduced by 70%, and the content of C content increased by 47%. Figure 15m shows the primer nanoclusters collected during laser paint removal (in ambient air, defocus distance +4 mm), which are composed of several nano-spheres generated by thermal ablation that attract each other and gather together. The content of iron was the least, only 0.52%. Compared to the primer particles collected during focused laser paint removal in ambient air (Figure 16g1), the iron content was reduced by 82.3%.

This is because the photons energy was absorbed by the paint and the substrate in the laser irradiation area and converted into heat during the extremely short time of laser pulse action, making it heat up and cool rapidly, with local rapid expansion and contraction within a short period of time. The absorbance, thermal conductivity, specific heat capacity and thermal expansion coefficient of the substrate and the primer are very different, resulting in a large difference in temperature and volume changes between the two and in considerable thermal stress and acceleration, so that the ablated primer particles were ejected from the sample together with part of the substrate metal. There were randomly distributed defects in the substrate. These were the places where the stress concentrated and the cracking occurred. Therefore, the surface of the side where the micron agglomerate (Figure 15g) detached from the substrate surface is rough and has small cracks. During paint removal in inert atmosphere, the content of the C element in the ablative products increased greatly, which was due to the strong air flow (Ar gas flow rate of 8–10 Lpm) blowing away the particles and smoke produced in time. The residence time of the ablated particles on the laser irradiation area decreased, which made most particles burn incompletely. Moreover, the content of Fe in the ablative products greatly decreased because there was no plasma, and violent physical and chemical phenomena such as splashing occurred during laser paint removal, which reduced the damage to the substrate.

## 4. Conclusions

The process combinations of laser defocus and open atmosphere (ambient air, compressed air and inert gas) were used to remove four-layer paint from the surface of an automobile’s body, which minimized the adverse effect on DC01 steel substrate. The main conclusions are as follows:The content of the C element and organic component in the samples laser treated (defocus distance +4 mm) in inert atmosphere was the lowest, and the organic component was reduced by 1.88% compared with the unpainted substrate. Their surface microtopography, roughness and hardness were very close to that of the unpainted substrate.With laser paint removal (in ambient air, defocusing distance 0 mm), the damage to the substrate is the most serious. The paint is removed at the focus of the laser; a remelted surface layer is formed on the surface of the substrate; the metallographic structure is changed; the crystal grains are refined, and the surface roughness is increased. The microhardness increases by more than 11 HV, and gradually decreases as the depth increases until it is 37 μm.In an open atmosphere (compressed air and inert gas, the gas flow is controlled at 8–10 Lpm) laser paint removal, only bright spots are seen when the paint layer is removed. Laser vaporization effects and thermal stress effects play a major role.In laser paint removal in an open atmosphere (compressed air and inert gas, the gas flow is controlled at 8–10 Lpm), the airflow can reduce local heat, make more energy act on the paint layer and accelerate the removal of the paint layer; by removing the ablation products, the generation of laser plasma and the shielding of the laser are avoided, and the damage of the substrate by the violent reaction is avoided. At the same time, the high-pressure gas blown during cleaning at the focal point can change the surface morphology and chemical composition content of the substrate, and the morphology and element content of the ablation product will also change.

## Figures and Tables

**Figure 1 materials-14-05969-f001:**
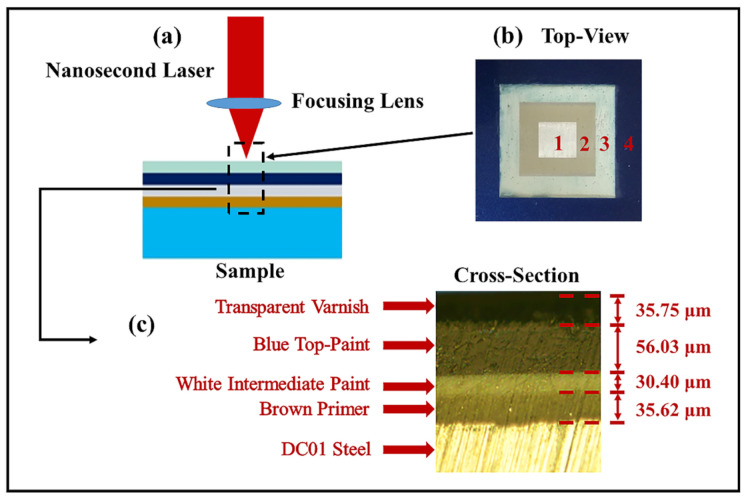
(**a**) Sample used for laser paint removal. (**b**) Top view of the sample. (**c**) Optical micrograph of cross-section of the sample.

**Figure 2 materials-14-05969-f002:**
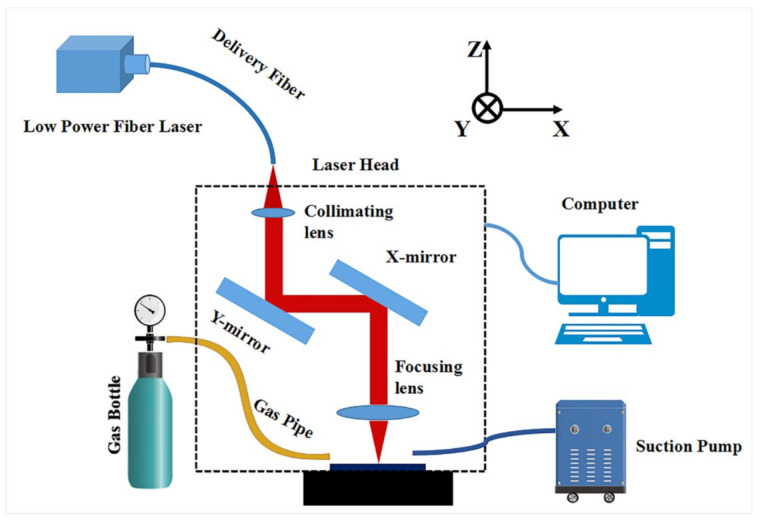
Schematic of setup for laser paint removal.

**Figure 3 materials-14-05969-f003:**
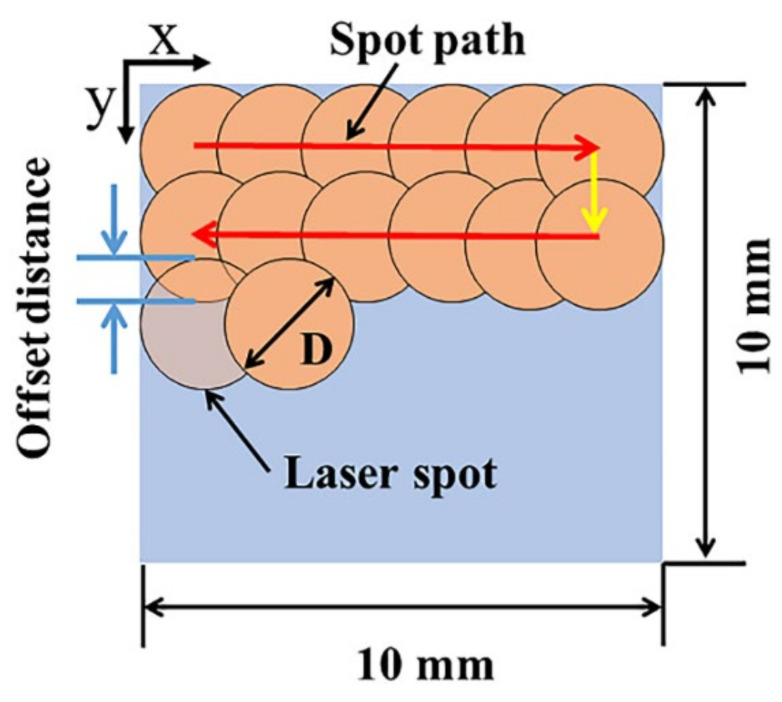
Schematic diagram of laser spot overlap.

**Figure 4 materials-14-05969-f004:**
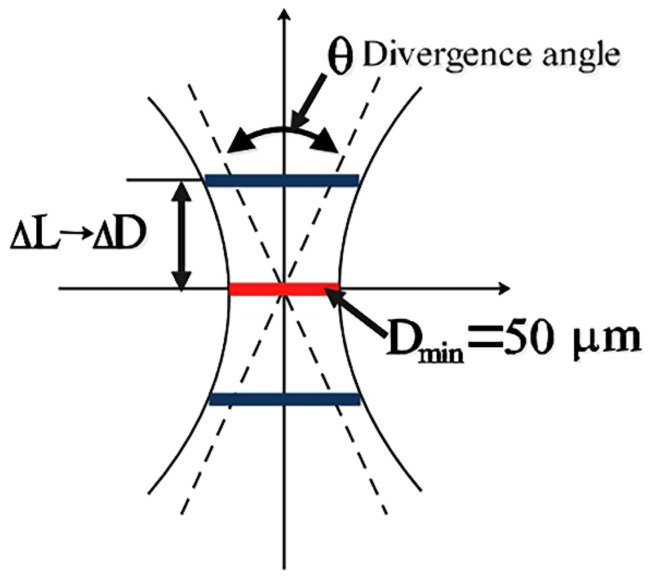
Schematic diagram of the size of light spot and defocus amount (Δ*L*: change in defocus amount, Δ*D*: change in spot diameter caused by change in defocus amount).

**Figure 5 materials-14-05969-f005:**
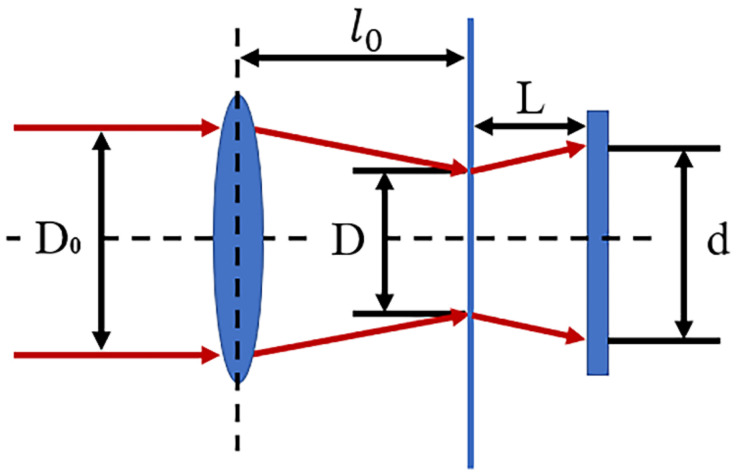
Defocus and spot size light path diagram. (D_0_: Incident laser beam diameter, D: Beam spot diameter, d: The laser beam actually acts on the sample spot diameter, L: Defocus, l_0_: Focal length).

**Figure 6 materials-14-05969-f006:**
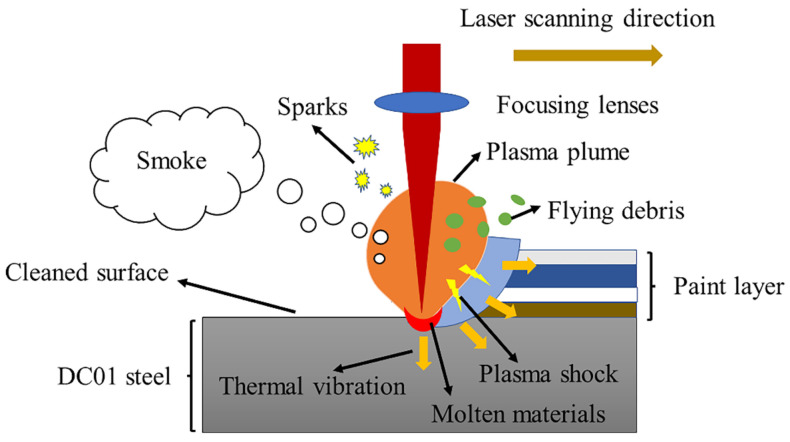
Schematic diagram of focused laser paint removal in ambient air.

**Figure 7 materials-14-05969-f007:**
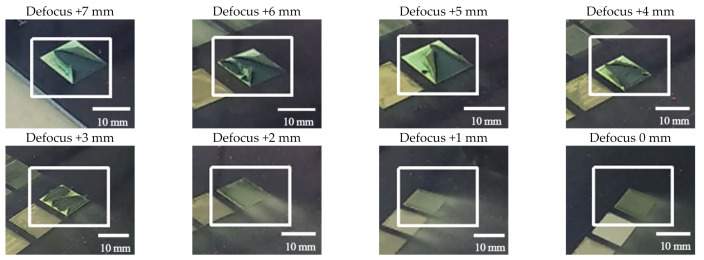
Photos of the samples after laser scanning once in ambient air when defocus distances were 0–+7 mm.

**Figure 8 materials-14-05969-f008:**
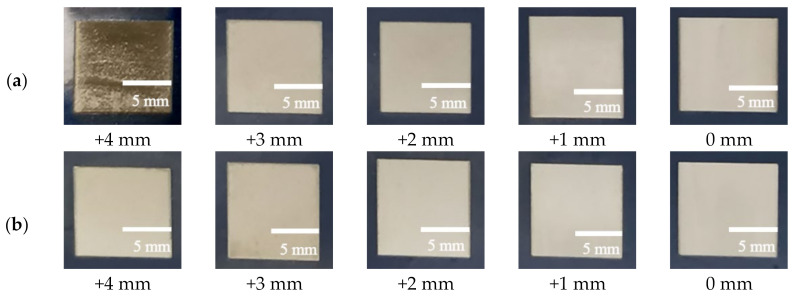
Photos of the samples laser scanned 3 times when defocus distances were 0–+4 mm: (**a**) in ambient air, (**b**) in inert gas (Ar).

**Figure 9 materials-14-05969-f009:**
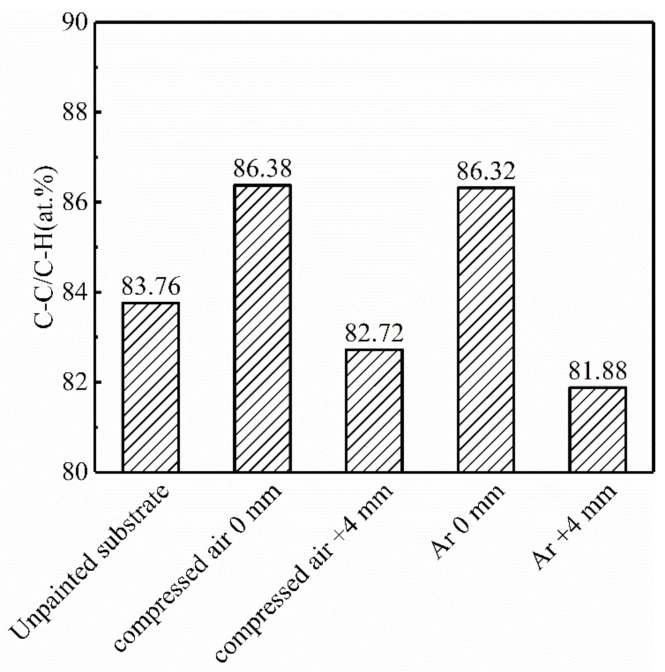
Relative amount of C-C/C-H functional groups (at.%) on samples measured by XPS.

**Figure 10 materials-14-05969-f010:**
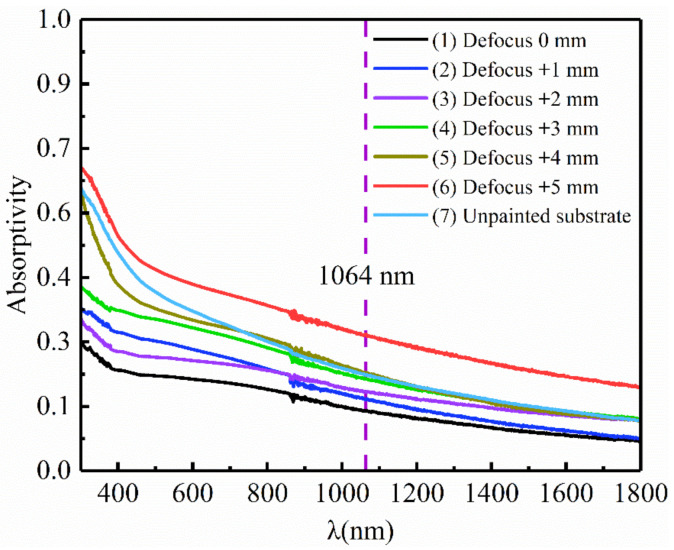
Absorbance of samples laser treated in inert atmosphere and unpainted substrate.

**Figure 11 materials-14-05969-f011:**
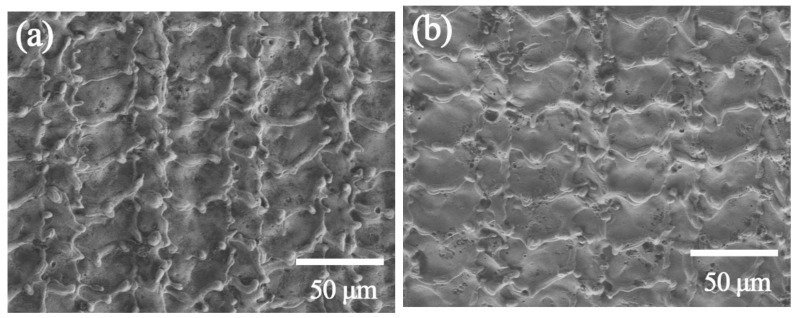
SEM images of laser treated samples: (**a**,**c**) top-view (ambient air, focused), (**e**,**e**-**1**) cross-section (ambient air, focused), (**b**,**d**) top-view (Ar atmosphere, focused), (**f**,**f**-**1**) cross section (Ar atmosphere, focused), (**g**) top-view (compressed air, defocus distance +4 mm), (**i**) cross-section (compressed air, defocus distance +4 mm), (**h**) top-view (Ar atmosphere, defocus distance +4 mm), (**j**) cross-section (Ar atmosphere, defocus distance +4 mm).

**Figure 12 materials-14-05969-f012:**
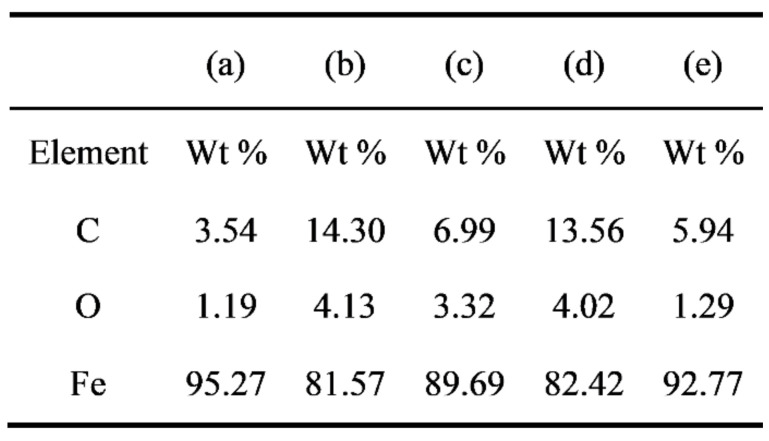
Representative EDS spectra of unpainted substrate and laser treated samples: (**a**) unpainted substrate; (**b**) in compressed air, focusing; (**c**) in compressed air, defocus distance +4 mm; (**d**) in inert atmosphere, focusing; (**e**) in inert atmosphere, defocus distance +4 mm.

**Figure 13 materials-14-05969-f013:**
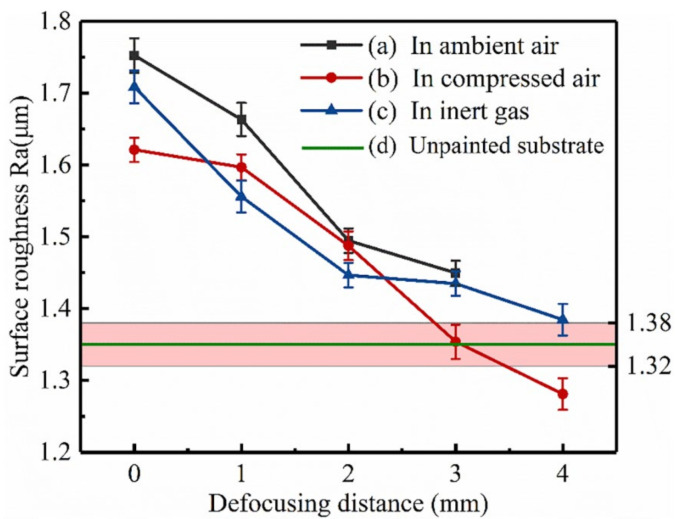
Surface roughness of samples laser scanned 3 times: (**a**) in ambient air, defocus distance 0–+3 mm; (**b**) in compressed air, defocus distance 0–+4 mm; (**c**) in inert atmosphere, defocus distance 0–+4 mm; (**d**) unpainted substrate.

**Figure 14 materials-14-05969-f014:**
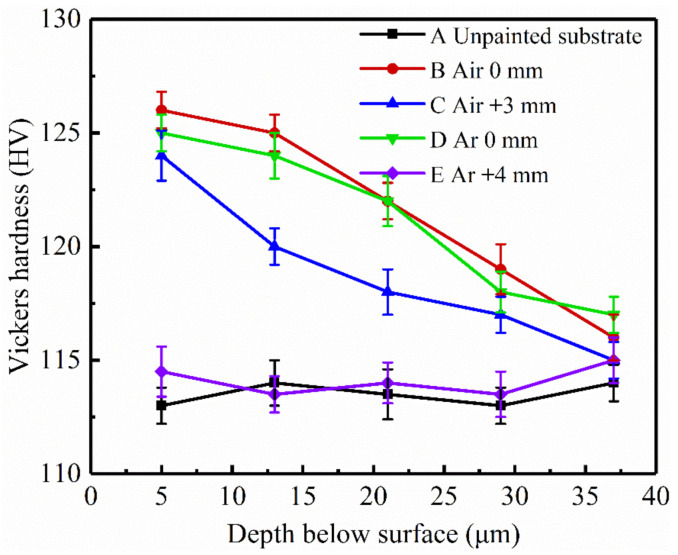
Microhardness depth profile of t unpainted substrate and laser treated samples: (**A**) unpainted substrate; (**B**) in ambient air, focused; (**C**) in ambient air, defocus distance +3 mm; (**D**) in inert atmosphere, focused; (**E**) in inert atmosphere, defocus distance +4 mm.

**Figure 15 materials-14-05969-f015:**
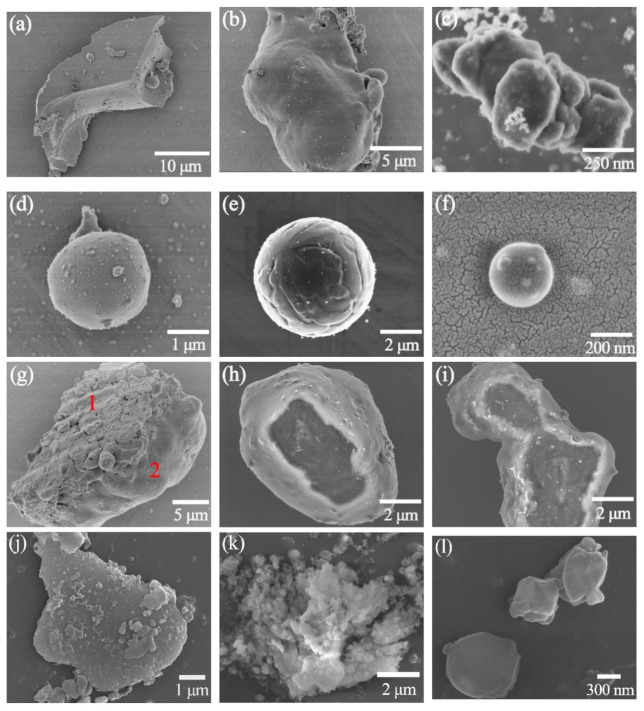
SEM images of particles collected by silicon wafer: (**a**–**g**) in ambient air, defocus distance 0 mm; (**h**) in ambient air, defocus distance +3 mm; (**i**–**l**) in inert atmosphere, defocus distance is 0 mm; (**m**–**o**) in inert atmosphere, defocus distance +4 mm.

**Figure 16 materials-14-05969-f016:**
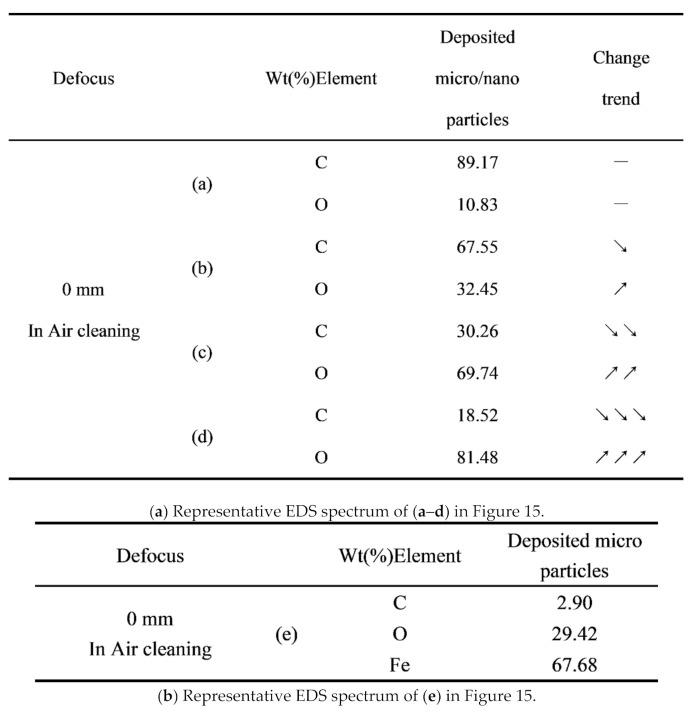
Representative EDS spectrum of (**a**–**e**), (**g**)1, (**g**)2, (**h**), (**i**), (**m**)1, (**m**)2 in Figure 15.

**Figure 17 materials-14-05969-f017:**
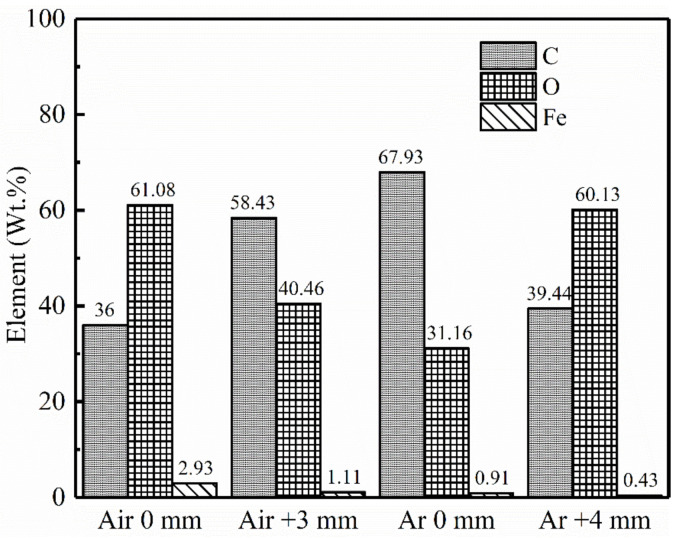
Content of C, O and Fe elements of particles collected by silicon wafers. 1st group: in ambient air, focusing; 2nd group: in ambient air, defocus distance +3 mm; 3rd group: in inert atmosphere, no defocusing; 4th group: in inert atmosphere, defocus distance +4 mm.

**Table 1 materials-14-05969-t001:** Components and physical properties of paint layers.

Paint Layers	Varnish	Colored Paint	Intermediate-Paint	Primer
Resin	Acrylic resin	Acrylic resin	Acrylic resin	Epoxy
Solvent	Aromatic hydrocarbons, Esters	Aromatic hydrocarbons, Esters	Aromatic hydrocarbons, Esters	Aromatic hydrocarbons, Esters
Additive	Leveling agent, Ehickener, Light stabilizer, etc.	Leveling agent, Dispersant, Light stabilizer, etc.	Leveling agent, Dispersant, Light stabilizer, etc.	Leveling agent, Dispersant, Light stabilizer, etc.
Pigment	None	Blue organic	Rutile titanium	Black organic
Colour	Transparent	Blue	White	Brown
vaporization temperature (°C)	108–144	>100	108–202	100

**Table 2 materials-14-05969-t002:** Laser processing parameters.

Characteristic	Value	Units
Wavelength	1064	nm
Maximum repetition frequency	1000	KHz
Maximum scanning speed	20,000	mm/s
Maximum average power	100	W
Maximum pulse energy	>1.0	mJ
Maximum peak power	>6	KW
Pulse width range	12–500	ns
Beam diameter	7.5	mm
M^2^ (Beam quality factor)	≤1.6	–

**Table 3 materials-14-05969-t003:** Laser processing parameters.

Laser Pass	Pulse Repetition Frequency (KHz)	Scan Speed (mm/s)	Spot Overlap Rate (%)	Pulse Width (ns)	Energy Density (J/cm^2^)
1st.	463	9260	60	100	10.9
2nd.	463	9260	60	100	10.9
3rd.	536	13,400	50	100	9.5

**Table 4 materials-14-05969-t004:** Spot diameter, fluence and overlap rate at different defocus surfaces.

Defocus Distance (mm)	Scan Times	0	+1	+2	+3	+4	+5	+6	+7
Spot diameter (mm)	-	0.05	0.079	0.109	0.138	0.167	0.197	0.226	0.255
Fluence (J/cm^2^)	1st 2nd	10.90	4.41	2.31	1.44	0.99	0.71	0.54	0.42
	3rd	9.51	3.81	2.00	1.25	0.85	0.61	0.47	0.37

**Table 5 materials-14-05969-t005:** Phenomenon during laser paint removal.

Atmosphere	Defocus Distance (mm)	Plasma	Debris Splash	Smoke	Flame	Dazzling Spot
air	0–+4	Yes	Yes	Yes	Yes	No
air	+5–+7	No	No	No	Yes	No
Compressed air	0–+7	No	No	No	No	Yes
Inert gas	0–+7	No	No	No	No	Yes

**Table 6 materials-14-05969-t006:** Absorbance of samples laser treated in inert atmosphere at wavelength 1064 nm.

Unpainted Substrate	Defocus 0 mm	Defocus +1 mm	Defocus +2 mm	Defocus +3 mm	Defocus +4 mm	Defocus +5 mm
0.212	0.133	0.159	0.176	0.204	0.218	0.303

## Data Availability

The data presented in this study are available on request from the corresponding author.

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
