# Peer review of "Effect of Defocused Nanosecond Laser Paint Removal on Mild Steel Substrate in Ambient Atmosphere"

_materials, 2021, doi:10.3390/ma14205969_

Round 1

Reviewer 1 Report

  1. The beam diameter or Gaussian beam radius on the sample is an important characteristic in laser structuring experiments because it can be used for the energy density (fluence) [J/cm2] characterization. The authors use a laser with a declared laser spot diameter of approximately 50 μm. Also, the authors declare table 4 with laser spot sizes and related influences at different focal positions. However, the evaluation method of spot size is not provided on paper. The reliable method for the evaluation of real spot sizes on the sample surface is the diameter squared versus fluence method [DOI: 10.1364/OL.7.000196; DOI: 10.1038/s41598-018-35604-z]. I would recommend including a technique description for measuring their beam size on the target material. Was it a CCD camera, or evaluation by equation (4)?
  2. The authors use a defocused laser beam for the paint layer removal. This type of experiment can be called a z-scan-type ablation experiment in which layer can be removed by the laser lift-off technique [DOI: 10.1364/OL.404760; DOI: 10.1002/adem.201901192] with the optimal removal condition (lowest removal threshold) approximately at defocus position equal to the Rayleigh length. What are optimal focusing conditions for paint removal in the presented work?
  3. Authors use nanosecond laser irradiation at 1064 nm of wavelength for 4 layer paint removal. It has been demonstrated the selection of the wavelength is important for the multi-layer system removal, because of absorptivity of each layer is dependent on the wavelength [DOI: 10.1016/j.phpro.2013.03.146; DOI: 10.1117/12.2212495]. What is the motivation for the selection of 1064nm wavelength, and how the different harmonics (532 nm, 355 nm, and 266 nm) would influence the paint removal?
  4. Authors use pulse overlap in their scanning procedure (different number of laser pulses per irradiation spot). It has been demonstrated the number of pulses per spot reduces the ablation threshold of layer removal because of heat accumulation [DOI: 10.1007/s00339-014-8255-0; DOI: 10.1051/epjap/2009029]. What are the author's opinions, is the bigger overlap and its related higher number of pulses per spot (smaller removal threshold) helps in paint removal or not?

Author Response

Your insightful comments have greatly improved the quality of the manuscript. The following pages are our point by point responses to your comments.

1.The beam diameter or Gaussian beam radius on the sample is an important characteristic in laser structuring experiments because it can be used for the energy density (fluence) [J/cm2] characterization. The authors use a laser with a declared laser spot diameter of approximately 50 μm. Also, the authors declare table 4 with laser spot sizes and related influences at different focal positions. However, the evaluation method of spot size is not provided on paper. The reliable method for the evaluation of real spot sizes on the sample surface is the diameter squared versus fluence method [DOI: 10.1364/OL.7.000196; DOI: 10.1038/s41598-018-35604-z]. I would recommend including a technique description for measuring their beam size on the target material. Was it a CCD camera, or evaluation by equation (4)?

reply:The beam diameter or Gaussian beam radius is evaluated by equation (4). Formula (4) is modified to formula (2).

2.The authors use a defocused laser beam for the paint layer removal. This type of experiment can be called a z-scan-type ablation experiment in which layer can be removed by the laser lift-off technique [DOI: 10.1364/OL.404760; DOI: 10.1002/adem.201901192] with the optimal removal condition (lowest removal threshold) approximately at defocus position equal to the Rayleigh length. What are optimal focusing conditions for paint removal in the presented work?

reply:In this work, in order to minimize the thermal impact of the laser beam on the substrate, the best defocus distance in this work is +4mm.

3.Authors use nanosecond laser irradiation at 1064 nm of wavelength for 4 layer paint removal. It has been demonstrated the selection of the wavelength is important for the multi-layer system removal, because of absorptivity of each layer is dependent on the wavelength [DOI: 10.1016/j.phpro.2013.03.146; DOI: 10.1117/12.2212495]. What is the motivation for the selection of 1064nm wavelength, and how the different harmonics (532 nm, 355 nm, and 266 nm) would influence the paint removal?

reply:The infrared fiber laser with a wavelength of 1064 nanometers was selected because of its good beam quality, very good monochromaticity, directivity and stability, and high conversion efficiency. In addition, this laser has the advantages of low cost, simple structure, and small footprint. Easy to use in large-scale and integrated industrial assembly lines. At the same time, we deleted the relevant content in Figure 11(a) after discussion.

4.Authors use pulse overlap in their scanning procedure (different number of laser pulses per irradiation spot). It has been demonstrated the number of pulses per spot reduces the ablation threshold of layer removal because of heat accumulation [DOI: 10.1007/s00339-014-8255-0; DOI: 10.1051/epjap/2009029]. What are the author's opinions, is the bigger overlap and its related higher number of pulses per spot (smaller removal threshold) helps in paint removal or not?

reply:In this experiment, a larger overlap of light spots within a certain range can remove more paint. So I chose a larger overlap in the first two paint removals.

Reviewer 2 Report

In the problem of laser removal of -paint coatings, two tasks are clearly separated- the study of the dependence on the power density of laser radiation and the dependence on the diameter of the laser beam (dimensional effect). The first allows us to highlight the role of various physical effects in the destruction of the paint layer (electron avalanche, ablation, acoustic destruction, etc.), the second is associated both with the features of various physical mechanisms of the primary destruction of the layer, and with the features of the removal of the products of destruction of coatings in a gas flow. In this case, it is most physical to study the first problem with a constant (or a number of constant) beam diameter, and the second problem with a constant power density.

In this paper, they investigate the defocusing method at one laser beam energy and duration. In this case, both the beam diameter and the power density change simultaneously. As a result, a hard-to-analyze mess of results is obtained, from which it can be and can be found a more or less optimal surface cleaning mode for a given type of coating, a given manufacturer and a given laser, but it is almost impossible to obtain specific physical results. This is reflected in the research results, which list all possible physical mechanisms known at this stage without special substantiation and quantitative interpretation. In this case, perhaps, new results are also lost. For ex. from the presented optical absorption spectra of coatings (which are not clear how they were obtained) it follows that a laser with a wavelength of ~ 1 μm is not optimal for these coatings.

The article uses the wrong formula and the corresponding figure (remarks 10 and 12), many inaccuracies, typos, inconsistencies in numerical parameters.

The article should not be published.

Remarks

  1. Lines 48-51 Repetition!
  2. Line 77 sealed CO2 slab Lase?

3.Line 96 50-micr-thick

  1. Lines 163,164. (lens focal length f = 254 mm) but not λ!
  2. Lines 173-176. What are gases purity?
  3. Fig. 2….. X, Y mirrors are not principal!
  4. Table 2 ……………………From the maximum pulse energy and pulse frequency the maximum power is 1000 W, not 100!
  5. Table 3 From Table 2 max Scan speed = 2000. Here 9260 -13400? What are diameters and energy of beams?
  6. Fig. 6 is not correct.
  7. Line  λ .: Focal length). Early Focal length = f.
  8. Line 224 Eq. 4 is not correct.
  9. Table 4 How does it coincide with tabl. 3?
  10. Fig. 7 Where is cleaned surface?
  11. Lines 326 -337. Were compressed gases checked on oil?
  12. 3.3. Surface absorptivity. How it was measured? What is substrate absorptivity? What are units?
  13. Line 576. LPM or lpm?
  14. Line 600. laser Vaporization
  15. Line 606. in time By removing

Author Response

Your insightful comments have greatly improved the quality of the manuscript. The following pages are our point by point responses to your comments.

Remarks

  1. Lines 48-51 Repetition!

reply:Duplicate part deleted

  1. Line 77 sealed CO2 slab Lase?

reply:already edited

  1. Line 96 50-micr-thick

reply:Modified to "50μm thick"

  1. Lines 163,164. (lens focal length f = 254 mm) but not λ!

reply:Lines 163,164. Modified to (lens focal length f = 254 mm)

  1. Lines 173-176. What are gases purity?

reply:The purity of inert gas is 99.5%.

  1. 2….. X, Y mirrors are not principal!

reply:This is just a schematic diagram of a laser processing device, to provide readers with reference.

  1. Table 2 ……………………From the maximum pulse energy and pulse frequency the maximum power is 1000 W, not 100!

reply:The laser power of 100W refers to the average output power. The maximum single pulse energy does not correspond to the maximum repetition frequency. Can’t use the two values to calculate the maximum power.

  1. Table 3 From Table 2 max Scan speed = 2000. Here 9260 -13400? What are diameters and energy of beams?

reply:Table 2 max Scan speed = 20000. The laser manufacturer did not provide this data.

  1. Fig. 6 is not correct.

reply:Figure 6 is derived from a simplified Gaussian laser beam

  1. Line λ .: Focal length). Early Focal length = f.

reply: Change to Line λ .: Focal length.

  1. Line 224 Eq. 4 is not correct.

reply:This formula is derived from the simplified theoretical calculation formula

  1. Table 4 How does it coincide with tabl. 3?

reply:The data in Table 3 corresponds to the data in the first column of Table 4

  1. Fig. 7 Where is cleaned surface?

reply:Modify Figure 7.

  1. Lines 326 -337. Were compressed gases checked on oil?

reply:Not checked.

  1. 3.3. Surface absorptivity. How it was measured? What is substrate absorptivity?

What are units?

reply:When testing, it is equipped with a blank control, which is automatically scanned and determined by the instrument near the specified wavelength. After measuring the absorbance of the sample, the instrument automatically deducts the blank reading and then calculates the absorbance. The absorbance of the substrate surface after laser cleaning and the absorbance of the original unpainted substrate surface. No unit.

  1. Line 576. LPM or lpm?

reply:Line 576. lpm

  1. Line 600. laser Vaporization

reply:Line 600. laser vaporization

  1. Line 606. in time By removing

reply:Line 606. in time by removing

Reviewer 3 Report

The manuscript entitled "Effect of Defocused Nanosecond Laser Paint removal on Mild Steel Substrate in Ambient Atmosphere" presents an interesting experimental study conducted on the removal of paint from the car parts by laser methods. However, multiple affirmations aren’t supported by the provided references or the experimental results obtained, and many other issues must be addressed. The paper needs minor revisions before it is processed further, some comments follow:

Some comments follow:

Introduction section

The introduction section must be improved. Multiple affirmations aren’t supported by the provided references or by the experimental results obtained. Please introduce citation at a specific position to assure a clear correspondence between the affirmations from the introduction section and the previous publication.

Please introduce corresponding citations for the following affirmations: "It usually consists…..  and a varnish (coat coat)."

Also, please remove the terms "(coat coat)" or replace them with a suitable term.

Could the author better explain the following sentence: "It has a wide range of applications." This is unclear, who or which method has a wide range of applications. Also, please introduce corresponding citations to previous studies that describe these methods.

The following sentence isn’t supported by the provided references: "However, the above methods have low precision, poor controllability, easy damage or contamination of the substrate, poor working environment, high labor intensity, and discharge of hazardous waste, require a large amount of flushing water, and are 38not environmentally friendly" Please introduce corresponding citations.

The following sentence was introduced twice: "These advantages.... are negligible." – please read carefully the entire manuscript. – remove the second sentence. Line 45-47 and line 48-51, correspondingly.

The following sentence is unclear: " By analyzing the treated.... optimal laser parameters were studied”. Please improve its clarity.

Line 105 - The following information is redundant: (Laser power density、Overlapping of neighboring scan lines) please remove.

Materials and Methods section

Line 146: The following sentence is redundant: "Painting is a mixture of resins, solvents, pigments and additives."

Table 1. How have been determined the characteristics of the paint layers?

Line 166-167: "The laser scanning system and the laser beam source were connected to the PC through the interface card," – please reduce the unnecessary information The method should be briefly described. Only the parameters and those data/information that are necessary for the experiment repeatability should be presented.

Figure 8 and Figure 9 – please introduce scale on the images.

Line 348 Please replace Fig. 10(a) with Fig. 11(a). Also please refer to the samples in the same way. The "original paint layer” sample isn’t presented in Figure 11.

The following sentence is unclear: "The absorbance. It can 346 be seen from the absorbance of the original paint layer, primer and original unpainted substrates in Fig.10(a) that the absorbance of the original paint layer, primer and original substrate at the laser wavelength of 1064 nm used in this work Respectively 0.508, 0.409, 349 and 0.212, the absorbance of the paint and the substrate is quite different, which is conducive to laser paint removal.”

Reference section

Please check carefully the correlation between the cited papers and the position of that reference in the manuscript text body. Some affirmations have no background in published literature.

Minor observations:

" 0.06 m2/min" and " CO2 " – please introduce superscripts or subscripts were necessary.

"slab Lase" – please check the manuscript for tipping errors.

"1455°C)”, "1.8mm” – please introduce a space between the values and the measuring units. Please make this correction in the entire manuscript.

"was 0~+2mm, and" – please remove the "+", please replace the symbol "~" with "÷" were appropriate.

I think that the study needs significant improvements

Author Response

Your insightful comments have greatly improved the quality of the manuscript. The following pages are our point by point responses to your comments.

Introduction section

Please introduce corresponding citations for the following affirmations: "It usually consists…..  and a varnish."

reply:After discussion, we will delete this sentence

Also, please remove the terms "(coat coat)" or replace them with a suitable term.

reply:"coat " replace them with a paint.

Could the author better explain the following sentence: "It has a wide range of applications." This is unclear, who or which method has a wide range of applications. Also, please introduce corresponding citations to previous studies that describe these methods.

reply:We will delete "It has a wide range of applications."

The following sentence isn’t supported by the provided references: "However, the above methods have low precision, poor controllability, easy damage or contamination of the substrate, poor working environment, high labor intensity, and discharge of hazardous waste, require a large amount of flushing water, and are 38not environmentally friendly" Please introduce corresponding citations.

reply:After careful discussion, we decided to delete the relevant literature

The following sentence was introduced twice: "These advantages.... are negligible." – please read carefully the entire manuscript. – remove the second sentence. Line 45-47 and line 48-51, correspondingly.

reply:remove the Line 45-47

The following sentence is unclear: " By analyzing the treated.... optimal laser parameters were studied”. Please improve its clarity.

reply:In order to eliminate the ambiguity we deleted this paragraph.

Line 105 - The following information is redundant: (Laser power density、Overlapping of neighboring scan lines) please remove.

reply:Line 105 (Laser power density、Overlapping of neighboring scan lines) remove.

Materials and Methods section

Line 146: The following sentence is redundant: "Painting is a mixture of resins, solvents, pigments and additives."

reply:Deleted line 146: "Painting is a mixture of resins, solvents, pigments and additives."

Table 1. How have been determined the characteristics of the paint layers?

reply:Table 1 The characteristics of the paint layer are determined by the paint characteristics data provided by the car manufacturer.

Line 166-167: "The laser scanning system and the laser beam source were connected to the PC through the interface card," – please reduce the unnecessary information The method should be briefly described. Only the parameters and those data/information that are necessary for the experiment repeatability should be presented.

reply:Deleted lines 166-167: "The laser scanning system and the laser beam source are connected to the PC through the interface card,"

Figure 8 and Figure 9 – please introduce scale on the images.

reply:The scale has been introduced in Figure 8 and Figure 9.

Line 348 Please replace Fig. 10(a) with Fig. 11(a). Also please refer to the samples in the same way. The "original paint layer” sample isn’t presented in Figure 11.

reply:Line 348 has replaced Figure 10(a) with Figure 11(a).

The following sentence is unclear: "The absorbance. It can 346 be seen from the absorbance of the original paint layer, primer and original unpainted substrates in Fig.10(a) that the absorbance of the original paint layer, primer and original substrate at the laser wavelength of 1064 nm used in this work Respectively 0.508, 0.409, 349 and 0.212, the absorbance of the paint and the substrate is quite different, which is conducive to laser paint removal.”

reply:We removed this part of the content from the article

Reference section

Please check carefully the correlation between the cited papers and the position of that reference in the manuscript text body. Some affirmations have no background in published literature.

Minor observations:

" 0.06 m2/min" and " CO2 " – please introduce superscripts or subscripts were necessary.

"slab Lase" – please check the manuscript for tipping errors.

reply:Superscripts or subscripts are used to distinguish the units and the correct use of chemical formulas, and to be consistent with the subscripts and subscripts elsewhere in the article. The minor error "slab Lase" in the manuscript has been changed.

Amend the sentence "slab Lase" to "The cleaning process was done with a single sealed CO2 laser (Rofin-Baasel, Multiscan VS). "

"1455°C)”, "1.8mm” – please introduce a space between the values and the measuring units. Please make this correction in the entire manuscript.

reply:In the entire manuscript, such as "1455°C)", "1.8mm"-a space has been added between the value and the measurement unit for correction.

"was 0~+2mm, and" – please remove the "+", please replace the symbol "~" with "÷" were appropriate.

reply:We don't think this part needs to be modified. "+" means the positive defocus amount, which is used to distinguish it from the negative defocus amount.

Round 2

Reviewer 2 Report

Unfortunately, the authors almost completely ignored the comments of the reviewer.

  1. On the experimental technique, separation of the main variables: power and beam diameter. It is clear that taking into account this remark requires additional measurements, time, which may be difficult for the authors. However,they can try to discuss such opportunities, to understand what additional valuable scientific information can be obtained.
  2. Spectral measurements. What do the authors mean by "absorptivity", especially for metal? What is this, skin effect? How are spectra measured for individual layers?
  3. The presence of oil in the compressed air. It is known that technical compressed air can contain noticeable traces of oil, which can lead to the appearance of a response of carbon atoms in the described experiments. The authors replied that they did not know about oil in compressed air.
  4. Using an incorrect description of Gaussian beams in focus. A correct description can be found, for example, in O. Svelto, Principles of Lasers, fifth ed., Springer N. Y., ISBN: 978-1-4419-1302-9,20 10. Or on the Internet. The presence of incorrect, approximate, moronic pattern Fig. 6 and formula Eq. 4 in a publication in a scientific international journal is unacceptable, especially since the article also contains a correct drawing. "The sun revolves around the Earth" - it can also be considered approximately!

Despite the large array of experimental data, this version of the article does not have high scientific value due to the unreliability of the results presented and gross errors. Publication is not advisable.

Author Response

Your insightful comments have greatly improved the quality of the manuscript. The following pages are our point by point responses to your comments.

  1. On the experimental technique, separation of the main variables: power and beam diameter. It is clear that taking into account this remark requires additional measurements, time, which may be difficult for the authors. However,they can try to discuss such opportunities, to understand what additional valuable scientific information can be obtained.

Reply: In this experiment, only the amount of defocus is changed, and the power of the laser has not changed, but the plane where the focus is located has changed relative to the plane to be cleaned. In this experiment, it is to explore the change of defocus amount and the influence of the use of auxiliary gas on the paint removal effect and related mechanisms.

  1. Spectral measurements. What do the authors mean by "absorptivity", especially for metal? What is this, skin effect? How are spectra measured for individual layers?

Reply: a change in surface absorptivity of a laser beam irradiation as a result of surface treatments was evaluated by the spectrometer, since the laser absorption is important to the consequent effect on the materials and it is dependent on the laser wavelength and material absorptivity. When laser light hits the surface of a solid metal, it may be absorbed in the metal with matched electromagnetic radiation. Absorption of laser light is the interaction of the electromagnetic radiation with the electrons of the material, this depends on the characteristics of the laser light and the properties of the metal. The absorptivity of a metal relates to its reflectivity.

  1. The presence of oil in the compressed air. It is known that technical compressed air can contain noticeable traces of oil, which can lead to the appearance of a response of carbon atoms in the described experiments. The authors replied that they did not know about oil in compressed air.

Reply: The trace oil in the compressed air will not have much influence on the test results. We have done a lot of repeated tests to get the results. The trace oil in the compressed air has little effect on the test results. The focus of our experimental test results is the amount of defocus and the use of auxiliary gas to affect the amount of residual organic matter on the surface after cleaning. A small amount of oil is not a serious defect in the entire experimental test result.

  1. Using an incorrect description of Gaussian beams in focus. A correct description can be found, for example, in O. Svelto, Principles of Lasers, fifth ed., Springer N. Y., ISBN: 978-1-4419-1302-9,20 10. Or on the Internet. The presence of incorrect, approximate, moronic pattern Fig. 6 and formula Eq. 4 in a publication in a scientific international journal is unacceptable, especially since the article also contains a correct drawing. "The sun revolves around the Earth" - it can also be considered approximately!

Despite the large array of experimental data, this version of the article does not have high scientific value due to the unreliability of the results presented and gross errors. Publication is not advisable.

Reply: Equation 4 is derived by Michael Bass. Laser Material Processing [M]. North-holland Publishing Company, 1983: 239-245. Figure 6 is based on the literature [doi:10.3390/coatings9080488],[DOI:10.3969/j.issn.1003501X.2017.03.009],[ https://doi.org/10.1007/s00339-020-03551-0], [DOI: 10.1007/s12206-009-0507-0], and the experimental phenomena and results of this experiment.

This manuscript is a resubmission of an earlier submission. The following is a list of the peer review reports and author responses from that submission.